# Crystallographic Features of Phase Transformations during the Continuous Cooling of a Ti6Al4V Alloy from the Single-Phase β-Region

**DOI:** 10.3390/ma15175840

**Published:** 2022-08-24

**Authors:** Inna A. Naschetnikova, Stepan I. Stepanov, Andrey A. Redikultsev, Valentin Yu. Yarkov, Maria A. Zorina, Mikhail L. Lobanov

**Affiliations:** 1Heat Treatment & Physics of Metals Department, Ural Federal University, 19 Mira Str., 620002 Ekaterinburg, Russia; 2M. N. Miheev Institute of Metal Physics, 18 S. Kovalevskaya Str., 620108 Ekaterinburg, Russia; 3Institute of Nuclear Materials, Sverdlovsk Region, 624250 Zarechny, Russia

**Keywords:** burgers orientation relationship, electron backscatter diffraction, phase transformation, titanium, shear transformation, variant selection

## Abstract

Crystallographic relationships between α- and β-phases resulting from phase transformations, which took place during the continuous water quenching (WQ), air cooling (AC) and furnace cooling (FC) of a Ti6Al4V plates solution treated at 1065 °C, were investigated by methods of electron backscatter diffraction (EBSD) and transmission electron microscopy (TEM). WQ, AC and FC resulted in typical martensite, basket-weave and parallel-plate Widmanstatten structures, respectively. The experimental distribution of α/β-misorientations deviated from BOR at set discrete angles close to 22, 30, 35 and 43°. The experimental spectra of angles were confirmed by theoretical calculations of the possible misorientations between the α and β phases through the β_o_→α→β_II_ –transformation path based on Burgers orientation relationship (BOR). Joint analysis of the experimental data and theoretical calculations revealed that the secondary β_II_-phase was precipitated according to the sequence β_o_→α→β_II_ during continuous cooling from the single-phase β-region. Similar spectra for α/β-phase misorientations for all investigated cooling rates acknowledged the similar transformation mechanisms and dominant shear component of the phase transformations.

## 1. Introduction

Ti-6Al-4V is a typical (α+β)-titanium alloy widely used in the aerospace, marine, chemical, and biomedical industries due to its high strength-to-weight ratio, good corrosion resistance and excellent stability of mechanical properties [1,2,3]. It frequently serves as a model alloy, i.e., to establish the effect of processing parameters, e.g., thermomechanical processing [4] and additive manufacturing [5] on structure evolution.

Polymorphic (allotropic) phase transformation in titanium and its alloys results in the formation of the product or daughter α-phase crystallographically linked with the parent or matrix β-phase. According to the Burgers orientation relationship (BOR): {0001}α || {011}β and <$11\bar{2}0$> α || <111>β [6], each α-orientation provides six β-variants during heating, while the β→α transformation results in the formation of 12 distinct α-variants from a single parent β-grain. Hence, one α-orientation can provide 72 different orientations during general solution treatment. However, one does not observe all 72 orientations of α-phase with equal probability in practice and the β→α-transformation takes place with variant selection [7,8,9,10,11,12,13,14].

Recently the peculiarities of α-variant selection have been thoroughly studied. It was established that the crystallographic characteristics of parent β-grain boundaries (misorientation and grain boundary plane orientation), interfacial energies [11,12,15] and the elastic strain energy of the transformation [9,13,16]) determine the α-variant, which has the lowest value of the nucleation barrier energy [11,17,18]. The variant selection in this case is based on the minimization of the interfacial energy between the daughter α-variant and two parent β-grains based on either the grain boundary plane [17,19] or the misorientation of two β-grains [12,20,21]. This variant selection mechanism is most common at low cooling rates. The second variant selection mechanism observed at high cooling rates is associated with self-accommodating three-variant clusters, which share a common <$11\bar{2}0$> direction to minimize the transformation strain [9,10,18,22,23]. It has also been demonstrated that there are special cases when α-grain boundaries follow BOR with only one prior β-grain and share a common basal plane with α on the non-Burgers-oriented side of the boundary. This happens when adjacent β grains with different orientations and secondary α-precipitates follow BOR in a particular β-grain. α-variants can nucleate following BOR with both prior β-grains if the β/β boundary shares a common (110)_β_ pole [11,23,24,25]. Despite the fact that numerous principles of variant selection have been stated, there are a number of studies [26,27] showing that some non-BOR α-β misorientations can be observed in titanium alloys. This could be due to that fact that β-phase precipitation evolution has received limited attention and still needs to be clarified. Therefore, the investigation of α/α and α/β-misorientations can contribute to the understanding of the interface and dislocation–grain boundary interaction [25,28,29] as well as the crystallographic texture evolution, which play key roles in mechanical properties and material performance [30,31,32,33,34,35]. 

Moreover, a number of studies have shown the effect of phase transformation paths [10,18,36], thermo- and thermomechanical processing [37,38,39], and chemical composition [40] on α/α grain boundary network characteristics (misorientation angle, plane orientation, population, connectivity) and on α precipitation mechanisms. Particularly, diffusion and shear are the two generally accepted phase transformation mechanisms in titanium alloys, which are usually controlled by varying the cooling rates from the single β-state [3] or (α + β)-field [41]. Higher cooling rates result in the shear mechanism; a slower cooling rate results in diffusion or a diffusion-assisted mechanism. Some authors have declared [42] that the features of both shear and diffusion mechanisms are typical of titanium alloys and high-alloyed steels. Other studies have described each transformation mechanism in detail and strictly divide diffusion and shear [8,10,22,43,44]. However, it has been demonstrated that the BOR is followed for all transformation mechanisms [18,23,43,45]. Though there are many theoretical approaches to classify the mechanisms of phase transformations, experimental observations of the crystallographic regularities can shed light on the theory of phase transformations [42,46].

Therefore, in the present work, hot-rolled plates of Ti-6Al-4V were solution-treated above β-transus (1065 °C) and water-quenched (WQ), air-cooled (AC), and furnace-cooled (FQ). A detailed characterization of the misorientations between α and β phases was made using EBSD and TEM and confirmed by the theoretical calculations for the β-α-β transformation path according to BOR. The formation of secondary β_II_ within the parent β_o_-grain according to the following route β_o_ →α →β_II_ was observed during cooling with various rates from the single-phase β-region. An assumption of the dominant role of shear in phase transformation during cooling from the β-state was made. Thus, this work contributes to the understanding of the crystallographic peculiarities of the phase transformations and enables predictions of the structural parameters and mechanisms of phase transformations, which define the ability to design and control the required properties of the metals and alloys.

## 2. Materials and Methods

β-transus temperature of 1033 °C for a hot-rolled Ti–6Al–4V alloy was obtained using differential scanning calorimetry on STA 449 Jupiter, Netzsch according to the method described in [47]. The as-received material was solution-treated at 1065 °C for 1 h and then water-quenched (WQ), air-cooled (AC) and furnace-cooled (FC), corresponding to cooling rates of ~100, 10, and 0.1 °C/s, respectively (Figure 1). 

The samples for EBSD were ground and polished with a DiaDuo diamond suspension followed by electropolishing in a chlorine-acetic electrolyte at an ambient temperature, a voltage of 40 V and a current of 5 A.

The orientation measurements were performed using a scanning electron microscope ThermoScience Scios 2 LoVac equipped with an Oxford Instrument Symmetry EBSD Detector. A scanning step was chosen from 0.05, 1 and 2 μm depending on the recognition quality of Kikuchi lines. Standard lattice parameters of the α (HCP) and β (BCC) phases were taken from the HKL database. The data were obtained using the Aztec software and processed using the HKL channel 5 software. To obtain the most reliable EBSD maps, a standard error cleaning procedure was carried out in accordance with the recommendations of Oxford Instruments. In addition, unclear data were processed by applying the single average orientation function.

The coordinate system (X, Y, Z) for the EBSD analysis was taken as follows: the X-axis was set to be parallel to the rolling direction (RD) and to the horizontal axis of the sample, the Y-axis was set to be parallel to the normal direction (ND) and to the vertical axis of the sample and the Z-axis was set to be parallel to the transverse direction (TD) and normal to the plane of the map. 

Misorientation angle distribution for α/α grain boundaries (GB) was calculated for all orientation maps. A part of the Oxford Instruments software, which analyzes the fulfillment and accuracy of the BOR between β- and α-phases and analyzes the interphase boundaries distribution according to the misorientation angles, was employed. The following BOR {$1\bar{1}0$}<111>β || {001}<110>α was set for the analysis. 

The reconstruction of the high-temperature β-phase was carried out using the AztecCrystals software according to the Burgers orientation relationship.

The TEM examination of thin foils of the Ti–6Al–4V alloy was conducted using a Talos F200X microscope operated at 200 kV.

To determine the deviations from BOR, the possible misorientations between α and β phases for the β→α→β transformation path were calculated by a matrix method similar to that described in [12,48].

## 3. Results and Discussion

The microstructure of the sample solution treated in single-phase β-region was characterized by a large parent β_o_-grain of the similar size for all cooling rates (Figure 2b,d,f). β-cooling in water, air and furnace leads to typical martensite, basket-weave and parallel-plate Widmanstatten structures [41]. Lamellar α(α’)-colonies of α-platelets of different thicknesses and orientations were metallographically distinguishable within the parent β-grains (Figure 2a,c) for WQ, AC and FC. Adjacent α-lamellas in the colony had only one common crystallographic element at higher cooling rates. In contrast, FC was characterized by a complete coincidence of the α-lamellas orientation within the colony (Figure 2e). The thickness of the α-lamellas in the colony increased with the decrease in the cooling rate. 

One can mention a difference in the morphology of the reconstructed high-temperature β_o_-grains. The β_o_-phase recrystallized during the treatment of the solution was generally characterized by large polyhedral grains with almost straight boundaries. This type of structure was reproduced for the high cooling rates (Figure 2b). Low-angle boundaries within the large β_o_-grains on the reconstructed maps were apparently related to the fact that phase transformations take place with some deviation from the Burgers OR. 

The size of the reconstructed β_o_-grains of the FC samples corresponded to those observed for other cooling rates (Figure 2f). However, the boundaries of reconstructed β_o_-grains were highly irregular: protrusions of reconstructed β_o_ grains noticeably penetrated into neighboring ones. This reconstruction error was due to the fact that the α-colony can simultaneously grow within two adjacent β_o_-grains with common crystallographic planes, as was mentioned in [24,27]. Thus, the reconstruction not only reflected the high-temperature β_o_-grain structure but also the features of α-phase nucleation and growth at high-angle boundaries during the β →α-phase transformation (Figure 2f). 

Relatively small areas of numerous secondary β_II_ precipitates with the orientation different from the matrix one were observed upon AC within parent β_o_-grains (Figure 2d). Some areas of β_II_ precipitates had a lamellar structure. Moreover, some β_o_-grains acquired two distinct orientations during reconstruction (Figure 2d marked with arrows). The boundaries of the reconstructed β_o_ grains in the case of air cooling were noticeably more serrated compared to the boundaries of the quenched samples. 

Various α-phase crystallographic orientations were separated mainly by high-angle boundaries, mostly with a misorientation angle close to 60 ± 5° (Figure 2e–g). There were also two noticeable peaks located close to 90 ± 2° and 10 ± 3° (Figure 2e–g). The α/α-misorientation angle distribution was essentially discrete and rather similar for all cooling rates. The spectrum of the misorientation angle distribution for any mechanism of β→α transformation points to the limited possibilities of α-phase variant selection, which was noted earlier in [2,3,4,5,6,7,8,9]. Note that according to [42], lamellar structures are observed when the structure is formed by the shear mechanism. 

Phase analysis revealed a minor, but sufficient for reliable identification, fraction of β-phase of about 1% for all cooling rates. The major fraction of the β-phase was observed at the boundaries of α-colonies and at the parent β_o_-grain boundaries. A small fraction of the β-phase was located within the α-colonies and was not attached to any high-angle boundaries (Figure 3a–c). The β-phase was identified both by color (white) and by interphase boundaries on phase maps. The black color characterized the deviation from BOR at the α/β interphase boundary at angles of more than 5°. The size of the β-phase appeared larger than it was due to the fact that the width of the α/β-interphase boundaries was set equal to three pixels for the better identification of precipitates and their distribution in the structure (Figure 3a–c). The low-angle boundaries (<10°, 1 pixel width) and high-angle boundaries (>10°, 2-pixel width) were given only for FC (Figure 3c).

The analysis of EBSD data for the α/β-misorientation for WQ sample demonstrated that it was mainly determined by the BOR and scattering did not exceed 4° (Figure 3d); however, some other peaks systematically appeared at the angles close to 22, 30, 35 and 43° in the plots of deviation from BOR (Figure 3d–f), which indicates the presence of four additional specific crystallographic orientations of the β-phase. For the FC sample, the BOR was almost exactly fulfilled; however, there was a tiny peak at the angle close to 30° (Figure 3f). The major fraction of crystallites for the AC sample corresponded to the orientations deviated from the BOR and the crystallites with exact BOR composed a minor fraction (Figure 3e). Note that the distribution of deviation from BOR was discrete. 

In a previous study [49], the identical distribution of deviation from the BOR was obtained for additively manufactured Ti-6Al-4V. However, the authors attributed the resulting deviations to the cyclic heating/cooling upon electron beam melting. In this study, the spectra of deviation from BOR was repeated for continuous cooling.

In the first approximation, the discreteness of the misorientation angle distribution could have been because several parent β_o_-grains with different orientations were found on the ESBD map. Thus, α-phase crystallites formed on both sides of the β-boundary can have non-Burgers orientations. This situation is typical for furnace cooling as can be seen from Figure 3c.

However, the analysis of the phase maps showed that the non-Burgers β-phase could be located within the parent β_o_-grain, mainly at the boundaries of α-colonies, which was confirmed by the misorientation analysis for the fragment map within individual parent β_o_-grains (Figure 4). Moreover, the peaks in the spectrum of deviation from BOR were located at the same specific angles close to 22, 30, 35 and 43° for all cooling rates. The analysis of the pole figures showed one strong orientation and several additional ones that had a strict crystallographic relationship with the strongest one: all orientations were connected by the same rotations around the same axes (Figure 4h,j,l). According to the evolution of the pole figures of the parent β_o_-grain after cooling, it was possible to track the orientations that appeared during the phase transformation (Figure 4g,i,k). The spectrum of deviation from BOR also had peaks near the angles close to 22, 30, 35, and 43° (Figure 4b,d,f). A similar analysis was carried out for each parent β_o_-grain for all cooling rates. The spectra of deviation from BOR and pole figures were qualitatively identical for all analyzed parent β_o_-grains.

It is important to notice that the misorientation angle distribution in the daughter α-phase within the parent β_o_-grain was repeated for all samples: the α-phase was also separated mainly by a set of high misorientation angle boundaries 60 ± 5°, 90 ± 2° and 11 ± 3° (Figure 4a,c,e). These angles were close to the expected misorientations of the α/α boundaries calculated from the BOR [27]. Thus, the phase transformations in the Ti-6Al-4V alloy obeyed BOR at any cooling rate. In addition, the set of discrete angles in the α/α-misorientation angle distribution were typical of any cooling rate but the intensity of the peak near 60 ± 5° decreased with an increasing cooling rate. The α/α-misorientation was mainly dominated by 60° boundaries at a high cooling rates. This was due to the fact that the formation of this boundary is most crystallographically favorable to minimize the transformation strain [39].

It was assumed that the observed β-phase could be a product of β_o_→α→β_II_ transformation, each stage of which obeyed BOR. β_II_ is a secondary phase with an orientation different from parent β_o_-phase. Hence, to calculate the observed deviations from the BOR between (α^i^) α-phase crystallites and secondary crystallites (β^ij^) within one parent β_o_-grain, the following relationships between the matrix definitions of α and β-phase lattices were used [49]: α^i^ = B_i_β_o_. Where B_i_ is the matrix of one of the variants of the BOR for the β→α transformation; β^ij^ = A_j_α^i^, where A_j_ is the matrix of one of the BOR for the reverse α→β transformation. 

The transition from any basis of secondary precipitates β_ij_ to any α^k^ basis can be rep-resented as α_k_ = B_i_A_j_A_k_^−1^β^ij^, where the set of matrices M = B_i_A_j_A_k_^−1^ describes the possible variants for the connection of the α-phase formed in one parent β_o_-grain according to the BOR with secondary β_II_-crystallites (β^ij^) also formed according to the BOR from the α^k^ orientations. Comparing all possible variants of the M matrix, while taking into account the symmetry of the β and α phases, with the matrix of one of the any variants of the BOR for β→α transformation, it was possible to obtain possible BOR deviation H matrices for primary α-crystallites and secondary β_II_- crystallites (β^ij^). For each H matrix, it was possible to calculate the values of the Θ-misorientation angles, taking into account the full range of its symmetrical options, according to the well-known mathematical formula: Θ = arccos{(h_11_ + h_22_ + h_33_ − 1)/2}, where h_ij_ are the elements of the H matrix. 

The calculation demonstrated that the experimentally obtained deviations from BOR at angles close to 0, 22, 30, 35 and 43° corresponded to BOR; however, such angles of deviation from BOR were typical of crystallites formed between the α-laths during the direct β_o_→α transformation and the β-phase formed during the reverse α→β_II_-transformation. In other words, these misorientations were not any new OR but were crystallographic equivalents of the BOR for β_o_→α→β_II_ transformation path.

The TEM studies revealed a typical mixture of elongated α-lamellas and interlamellar β-phase. The selected electron diffraction patterns show reflections from the α-phases and β-phase (Figure 5). Based on the obtained indexes of the zone axis—[$1\bar{1}0$] for the α-phase and [001] for the β-phase, the angles of deviation from BOR for α-and β-phases were calculated. The angles were 36.83° for both α_1_/β and α_2_/β variants, which was experimentally observed to deviate from the BOR angle distribution according to ESBD (Figure 3d and Figure 4b,d).

Although the minor fraction of the β_II_-phase deviated from BOR was observed for the FC phase maps obtained with low magnification, the α/β-misorientation at higher magnifications analysis revealed that β_II_ could be observed locally (Figure 6 and Table 1). Such analysis enabled five groups to be categorized by the sites of β_II_-precipitation and crystallographic misorientation with an adjacent α-phase: between the α-laths of the same orientation in one α-colony, 0° deviation from BOR;between the α-laths of the same orientation in one α-colony, deviation from BOR at one of the angles close to 22, 30, 35 and 43°;at a high-angle boundary (mainly α1/α2-misorientation close to 60°), 0° deviation from BOR with both α1 and α2;at a high-angle boundary (mainly α1/α2-misorientation close to 60°), 0° deviation from BOR with α1 and deviation from BOR at one of the angles close to 22, 30, 35 and 43° with α2;at a high-angle boundary (mainly α1/α2-misorientation close to 60°), deviation from BOR at one of the angles close to 22, 30, 35 and 43° with both α1 and α2.

The transformation of the crystallite orientation in the material can occur upon heating and/or deformation processing. Since the samples were not subjected to such influence during cooling below the β-transus, the appearance of additional β-phase orientations, which were crystallographically different from the high-temperature β_o_, can be explained by the secondary α→β_II_-precipitation after the β_o_→α-transformation. Such precipitation is thermodynamically possible for an alloy cooled in the (α + β)-region. Thus, the occurrence of peaks in the range of 22–43° in the spectrum of deviation from BOR (Figure 3d–f and Figure 4b,d,f) points to the possibility of the precipitation of the β_II_-phase from α-phase upon cooling. Moreover, there were peaks at the same angles in all considered spectra of deviation from BOR, i.e., the secondary β_II_-phase precipitation following BOR took place for all cooling rates. 

The same morphology of the secondary β_II_-phase precipitates, the sites of its precipitation, and the symmetry of orientations (according to the PF) give reason to believe that these precipitations had a single formation mechanism. The obeying of the BOR for α→β_II_, the sites of β_II_-precipitation, discrete angles of deviation from BOR, along with the lamellar morphology of the α-phase and the discreteness of the α/α-misorientation angle distribution implies a significant contribution of the shear component to of β_o_→α→β_II_-phase transformation for all the considered cooling rates.

## 4. Conclusions

The analysis of the misorientations between α/α-crystallites and α/β-phases was performed for Ti-6Al-4V subjected to super transus solution treatment and continuous water, air and furnace cooling. The precipitation of the secondary β_II_-phase, crystallographically different from the high temperature β_o_-phase was characterized by deviation from the BOR at a discrete set of the angles close to 22, 30, 35 and 43°. Theoretical calculations of α/β-misorientation based on BOR through β_o_→α→β_II_-transformation path confirmed the experimental set of angles. The frequency of the angles distribution depended on the cooling rate. The major deviation from BOR at angles close to 22, 30, 35 and 43° was typical of AC samples compared to WQ and FC samples. The precise crystallographic inheritance in the course of the β_o_→α→β_II_-transformation allows the conclusion that there was a dominant shear component in the mechanism of phase transformations during the cooling of the Ti-6Al-4V from the single-phase β-region regardless of the cooling rate.

## Figures and Tables

**Figure 1 materials-15-05840-f001:**
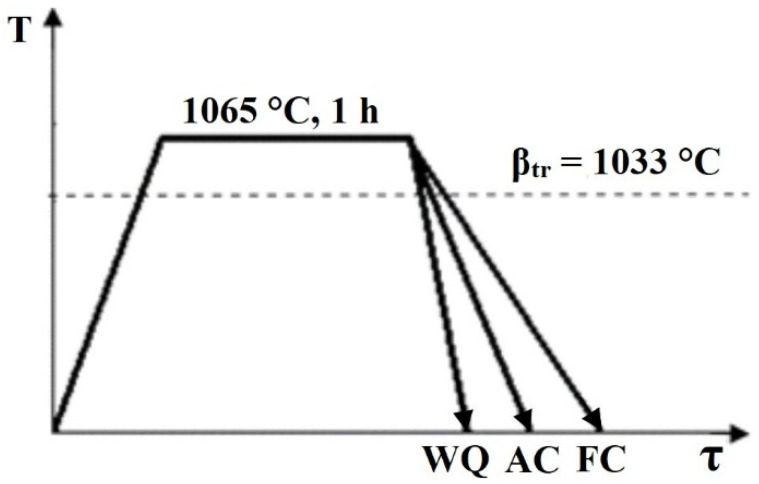
Heat treatment procedure for solution-treated Ti-6Al-4V, where WQ is water quenching, AC—air cooling and FC—furnace cooling.

**Figure 2 materials-15-05840-f002:**
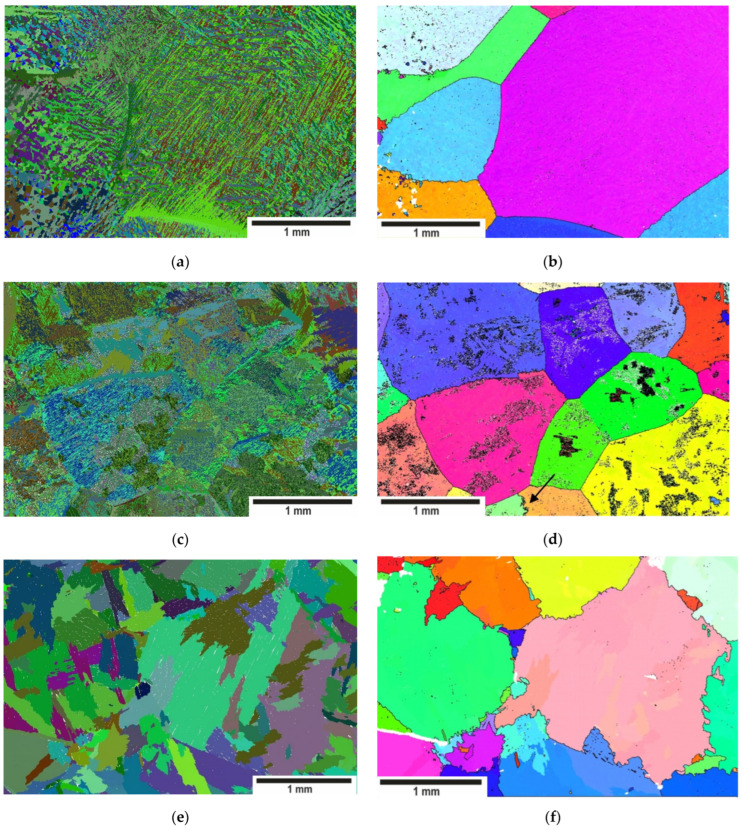
The orientation maps for the samples depending on the cooling rate in the form of orientation maps: (**a**) WQ, (**c**) AC, (**e**) FC; the orientation maps of the reconstructed high-temperature β-grain: (**b**) WQ, (**d**) AC, (**f**) FC; the misorientation angle distribution in the α-phase for WQ, AC, FQ: (**g**) WQ, (**h**) AC, (**i**) FC; (**j**) IPF color key.

**Figure 3 materials-15-05840-f003:**
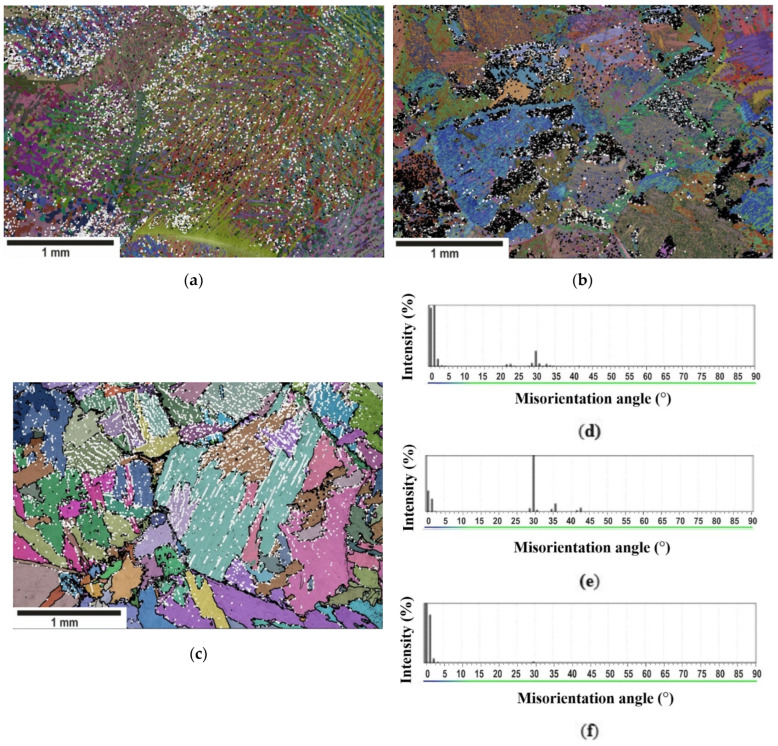
Phase maps with the identification of BOR at the α-β-interphase boundaries for: (**a**) WQ, (**b**) AC, (**c**) FC samples (α-phase—superimposed band contrast and Euler angles; β-phase—white; black color characterizes the α/β-boundaries deviated from BOR); Spectra of deviation from BOR for: (**d**) WQ, (**e**) AC, (**f**) FC samples.

**Figure 4 materials-15-05840-f004:**
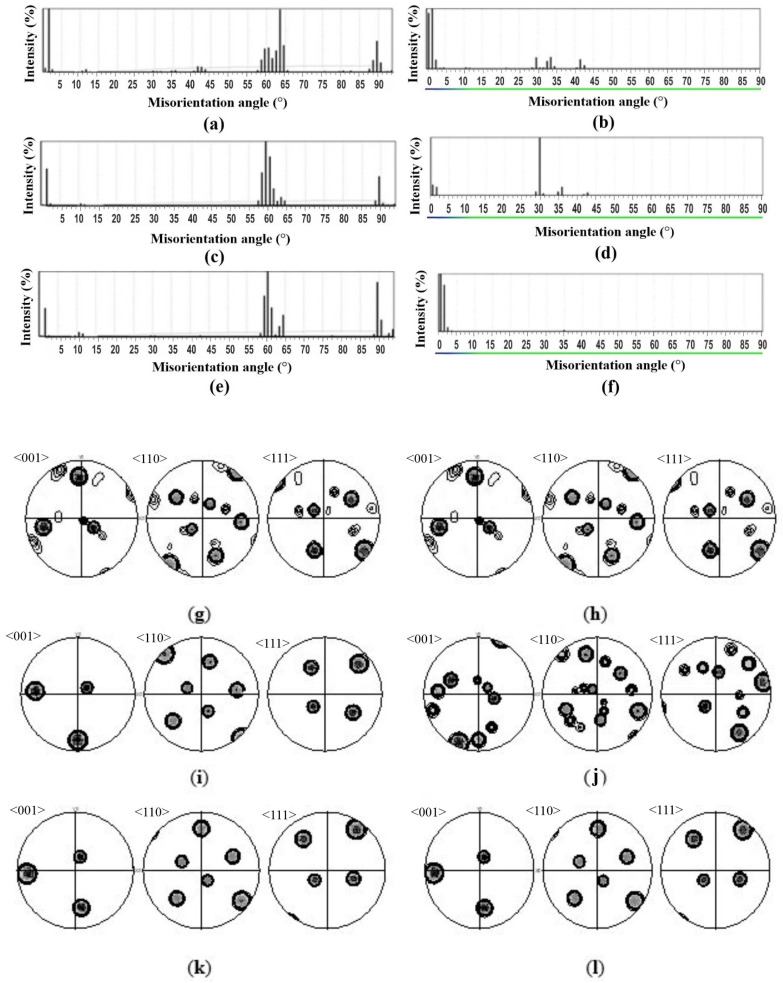
α/α and α/β misorientation analysis within one parent β_o_-grain depending on the cooling rate: (**a**,**c**,**e**) the misorientation angle distribution in the α-phase for WQ, AC, FC; (**b**,**d**,**f**) spectra of deviation from BOR for WQ, AC, FC; (**g**,**i**,**k**) <001>, <110>, <111> pole figures of the reconstructed β_o_-phase for WQ, AC, FC; (**h**,**j**,**l**) <001>, <110>, <111> pole figures of β-phase after WQ, AC, FC.

**Figure 5 materials-15-05840-f005:**
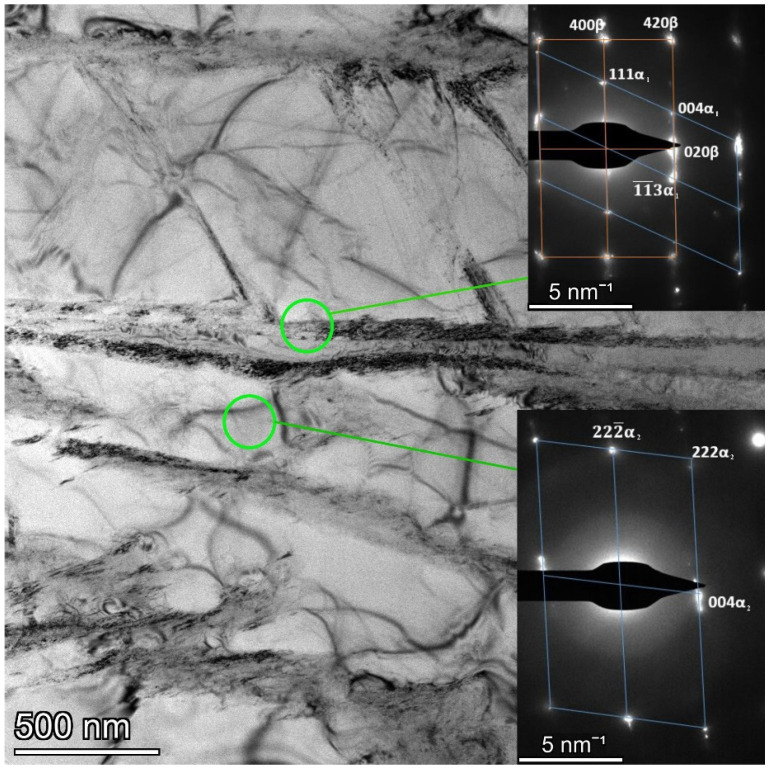
TEM bright field image of the AC sample and the corresponding selected area electron diffraction patterns.

**Figure 6 materials-15-05840-f006:**
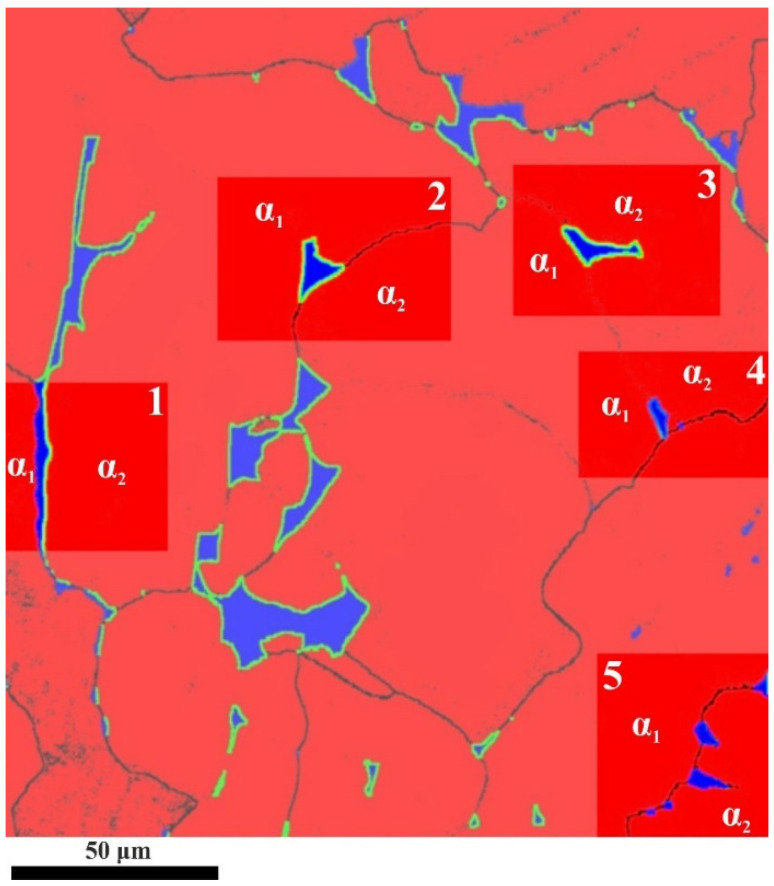
Phase map for the solution-treated and FC Ti-6Al-4V showing β-phase precipitation variants; α-phase—red; β-phase—blue; green color characterizes the deviation from BOR at angles > 5°.

**Table 1 materials-15-05840-t001:** Grain boundaries misorientations in Figure 6.

Section No.	α_1_/α_2_ Misorientation	BOR Deviation α_1_/β	BOR Deviation α_2_/β
1	67°	0°	34°
2	48°	34°	29°
3	3°	29°	29°
4	3°	0°	0°
5	61°	0°	0°

## Data Availability

Not applicable.

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
