# Peer review of "Crystallographic Features of Phase Transformations during the Continuous Cooling of a Ti6Al4V Alloy from the Single-Phase β-Region"

_materials, 2022, doi:10.3390/ma15175840_

Round 1

Reviewer 1 Report

The phase transformation of Ti6Al4V during continuous water quenching (WQ), air cooling (AC) and furnace cooling (FQ) after solution-treated at 1065°Ð¡ was investigated using electron backscatter diffraction (EBSD) and transmission electron microscopy (TEM) to explore the crystallographic relationship between the α and β phases. The possible misorientation angle distributions between the α - β phases were calculated based on Burgers orientation relations (BOR). The reported finding is interesting and can be considered for publication after some mandatory revisions.

-Keywords are redundant and need to be refined.

-Three cooling methods have been used; the differences caused by various cooling rates should be better clarified, particularly in abstract and conclusions.

-In Figure 2, the scanned regions are too small for texture interpretation.

-The horizontal and vertical coordinates of the misorientation angle distribution should be clearly indicated.

-Although the authors believe that the β→α transformation in this alloy always proceeds via a shear mechanism at various cooling rates, the specific effects of cooling rates should still be discussed.

-Several recent publications addressing very close topics to your work should be well referenced and discussed: doi.org/10.1007/s11661-022-06737-8; doi.org/10.1007/s10853-020-04603-9.

Author Response

Thank you for your time and valuable recommendations. We performed the corresponding edits to the Keywords and Figures, clarified the differences between the cooling rates, extended the reference list and answered all issues point-by-point.

All corrections are marked in the manuscript by blue highlighting.

Comment

Reply

1.   Keywords are redundant and need to be refined.

Please find the corresponding edits in the revised Kewords list.

2.   Three cooling methods have been used; the differences caused by various cooling rates should be better clarified, particularly in abstract and conclusions.

Please find the corresponding edits in the revised manuscript: Abstract, Lines 15-16; Conclusion, Lines 357-359

3.   In Figure 2, the scanned regions are too small for texture interpretation.

The purpose of the study was to investigate the crystallographic misorientations between α/α and α/β crystallites. For such analysis, more than 2 parent β-grains are enough. We used 5 and more.

The integral texture analysis was not implied, though the peculiarities of α/α and α/β-misoreintation formation can contribute to principles of texture formation.

4.   The horizontal and vertical coordinates of the misorientation angle distribution should be clearly indicated.

Please find the corresponding edits in Fig. 2.

5.   Although the authors believe that the β→α transformation in this alloy always proceeds via a shear mechanism at various cooling rates, the specific effects of cooling rates should still be discussed.

We have reworked the first Paragraph of Results and Discussion. The effect of cooling rate on structure of α-phase is now introduced.

6.   Several recent publications addressing very close topics to your work should be well referenced and discussed: doi.org/10.1007/s11661-022-06737-8; doi.org/10.1007/s10853-020-04603-9.

Thank you for these valuable references. We have cited and discussed them.

Reviewer 2 Report

In the manuscript entitled "Crystallographic features of phase transformations during continuous cooling of Ti6Al4V alloy from single-phase β-region " by  Inna et al., the authors report the phase transformations during continuous cooling of Ti6Al4V. It contains some interesting results; however, there are several lacks in the manuscript that need to be solved.

The manuscript has major issues, I do have concerns:

11. Please explain the novelty of the current work. The reviewer is not able to understand how this paper adds value to the scientific literature.

22. Introduction should include the most recent literature. The literature should have a mix of authors from different countries. The authors seem to have cited literature only from a particular country (more than 75% ).

33. Please explain with experimental results how the β-transus temperature of 1033°C was obtained.

44. Fig. 1: Please include arrow heads.

 5. Many results are explained in methodology section. They have to be taken to results and discussion section.

66.  Fig 4: The crystallographic planes are faintly visible. Please make it readable.

77.   The discussion should be strengthened.

Author Response

Reviewer â„–2 comments to authors

Thank you for your time and valuable recommendations. We performed reworked Introduction to emphasize novelty of the study, as well as Material and Methods according to comments, extended discussion and answered all issues point-by-point.

All corrections are marked in the manuscript by blue highlighting.

Comment

Reply

1.         Please explain the novelty of the current work. The reviewer is not able to understand how this paper adds value to the scientific literature.

We reworked the Introduction and now the novelty is given in the final paragraph - Lines 83-91.

2.         Introduction should include the most recent literature. The literature should have a mix of authors from different countries. The authors seem to have cited literature only from a particular country (more than 75% ).

We have no national or country preferences. We initially had introduced the literature review based on current achievements in the research topic and now expanded it in accordance with reviewers’ comments.

3.         Please explain with experimental results how the β-transus temperature of 1033°C was obtained.

We have pasted the device used for obtaining the β-transus - STA 449 Jupiter, Netzsch. The method based on the analysis of differential scanning calorimetry curves is described in detail in [doi:10.4028/www.scientific.net/SSP.284.259]. The reference is given in the manuscript. 

4.         Fig. 1: Please include arrow heads.

Please find the corresponding edits in Fig. 1 of the manuscript.

5.         Many results are explained in methodology section. They have to be taken to results and discussion section.

We moved the theoretical calculations of α/β-misorientations from Methods to Results and Discussion.

6.         Fig 4: The crystallographic planes are faintly visible. Please make it readable.

Please find the corresponding edits in Fig. 4 in the revised manuscript

7.         The discussion should be strengthened.

We have expanded the Discussion section. Please find the corresponding edits in the revised manuscript.

Reviewer 3 Report

The variant selection during different cooling rates from the high-temperature beta phase is investigated using various electron microscopy techniques. The work then compares the experimental results with their theoretical calculations to reveal the possible formation of a secondary beta phase during the different cooling procedures. The work is an excellent fundamental study that can be considered a reference for other processes, especially many additive manufacturing techniques where different cooling rates are experienced. However, there are minor issues that need to be addressed before its publication:

1)      The text needs to be revisited in terms of English grammar. Many long sentences need to be shortened and simplified (the first sentence of the introduction, for example).

2)      At the start of the introduction, “solid metals and alloys results” what do you mean by solid metals? It is better to use simple phrases such as “phase transformation in metals”.

3)      The experimental section: line 121-124à the sentence needs to be shortened. Also, what does it mean “the relationship between phase lattices were used”? Do you mean BOR?

4)      Also, lines 126-132: what are the Ci matrices? It hasn’t been mentioned in the previous paragraphs. A proper relationship between the Bi and Ci matrices is needed in the text.

5)      Results and discussion Figure 2: the IPF colour code related to the hcp and bcc crystals needs to be added to the Figure.

6)      To discuss the serrated boundaries, you need to add the boundaries to the IPF images of the reconstructed beta grains. This is currently missing in Figures 2b, d and f.

7)      Line 180-181 why the shear component of the phase transformation can be identified through the MAD? The MAD is similar for all transformation routes due to the crystallographic constraints of the transformation, though not necessarily pointing to the transformation mechanism.

8)      The distribution of the beta phase in the different cooling rates can be distinguished only in Figure 3c. You may need to add the band contrast image to the phase analysis image to prove your point in lines 190-197.

9)      How are the invariant boundaries indexed in Figure 3? How can it be identified as a separate phase? Clear information is required.

10)   Why the secondary beta cannot be considered as the primary beta grains that haven’t been transformed back into alpha during the cooling? In this case, we will also end up with already transformed alpha from one prior beta grain and another beta grain that hasn’t been transformed to alpha during the cooling cycle. This means that there would also be prior alpha/beta boundaries that deviate from the BOR. Clear explanations are needed to describe the phenomenon better.

Author Response

Reviewer â„–3 comments to authors

Thank you for your time and valuable recommendations. We performed the corresponding edits to figures and text, clarified why the β-phase is secondary and answered all issues point-by-point.

All corrections are marked in the manuscript by blue highlighting.

Comment

Reply

1.         The text needs to be revisited in terms of English grammar. Many long sentences need to be shortened and simplified (the first sentence of the introduction, for example).

Please find the corresponding edits in the revised manuscript

2.         At the start of the introduction, “solid metals and alloys results” what do you mean by solid metals? It is better to use simple phrases such as “phase transformation in metals”.

We meant solid, i.e. not liquid. We reworked the Introduction. Please find the corresponding edits.

3.         The experimental section: line 121-124 the sentence needs to be shortened. Also, what does it mean “the relationship between phase lattices were used”? Do you mean BOR?

This analysis was included in Results and Discussion section according to another Reviewer recommendation. According the comment 3 the description was shortened. Ci was excluded from the manuscript in order to make it clear to readers.

4.         Also, lines 126-132: what are the Ci matrices? It hasn’t been mentioned in the previous paragraphs. A proper relationship between the Bi and Ci matrices is needed in the text.

5.         Results and discussion Figure 2: the IPF colour code related to the hcp and bcc crystals needs to be added to the Figure.

We have added the IPF colour code for BCC in Figure 2. For HCP, the Euler angles maps are used. 

6.         To discuss the serrated boundaries, you need to add the boundaries to the IPF images of the reconstructed beta grains. This is currently missing in Figures 2b, d and f.

Please find the corresponding edits in in Figures 2b, d and f in the revised manuscript.

7.         Line 180-181 why the shear component of the phase transformation can be identified through the MAD? The MAD is similar for all transformation routes due to the crystallographic constraints of the transformation, though not necessarily pointing to the transformation mechanism.

We agree with Reviewer comment. The shear component of the phase transformation can’t be identified through the MAD. We’ve excluded the phrase from discussion.

8.         The distribution of the beta phase in the different cooling rates can be distinguished only in Figure 3c. You may need to add the band contrast image to the phase analysis image to prove your point in lines 190-197.

The amount of high-angle boundaries in α-phase is so great for WQ and AC that makes the map unrecognizable.

In revised version, to visualize the α-phase structure we superimposed band contrast, Euler angles and phase maps to demonstrate distribution of the β-phase. BOR β-phase now coloured white and has white boundaries. β-phase deviated from BOR at angles 22, 30, 35, and 43° has black boundaries.

The corresponding edits are made in the manuscript.

9.         How are the invariant boundaries indexed in Figure 3? How can it be identified as a separate phase? Clear information is required.

10.     Why the secondary beta cannot be considered as the primary beta grains that haven’t been transformed back into alpha during the cooling? In this case, we will also end up with already transformed alpha from one prior beta grain and another beta grain that hasn’t been transformed to alpha during the cooling cycle. This means that there would also be prior alpha/beta boundaries that deviate from the BOR. Clear explanations are needed to describe the phenomenon better.

We describe this case as a primary hypothesis in Lines 203-205. However, further detailed analysis of fragments within the individual parent β-grain shows that there is β-phase deviated from BOR in single parent β-grain (Fig. 4). Moreover, we demonstrate case, when such β-phase can precipitate not only between α-colonies of the same parent β-grain but also even between α-laths of the same orientations (Figure 6, Lines 286-289).

Round 2

Reviewer 1 Report

Now the manuscript can be accepted.

Reviewer 2 Report

The authors have minimally tried to address all the comments. I therefore recommend to accept it.